# Acute cardiovascular responses of postmenopausal women to resistance training sessions differing in set configuration: A study protocol for a crossover trial

Iván Nine[1], Alexis Padrón-Cabo[1,2], Eduardo Carballeira[1], Jessica Rial-Vázquez[1,3], María Rúa-Alonso[1,4,5], Juan Fariñas[1], Manuel Giráldez-García[1], Eliseo Iglesias-Soler[1] *

1 Performance and Health Group, Department of Physical Education and Sport, Faculty of Sports Sciences and Physical Education, University of A Coruna, A Coruña, Spain, 2 Faculty of Education and Sport Sciences, University of Vigo, Pontevedra, Spain, 3 Centre for Sport Studies, Rey Juan Carlos University, Fuenlabrada, Madrid, Spain, 4 Research Center in Sports Sciences, Health Sciences, and Human Development, Vila Real, Portugal, 5 Polytechnic of Guarda, Guarda, Portugal

* eliseo.iglesias.soler@udc.es

**Editor:** Daniel Boullosa, Universidad de León Facultad de la Ciencias de la Actividad Física y el Deporte: Universidad de Leon Facultad de la Ciencias de la Actividad Fisica y el Deporte, SPAIN

## Abstract

### Background

Resistance training is hardly recommended for postmenopausal women to counteract negative effects of hormonal changes. However, some concern exists about the marked hemodynamic responses caused by high-load resistance exercises. In this regard, studies on young, healthy, physically active individuals suggest that set configuration can modulate acute cardiovascular, metabolic, and cardiac autonomic responses caused by resistance training sessions, but this has not yet been explored in postmenopausal women.

### Methods

A sample of 60 physically active postmenopausal women (30 normotensive, 30 hypertensive) will participate in this crossover study. After a medical exam, ergometry, familiarization session, and two testing sessions, participants will complete three experimental sessions and one control session in a randomized order. Each experimental session includes 36 repetitions of four exercises (horizontal leg press, bench press, prone leg curl, and lat pulldown) differing in set configuration: 9 sets of 4 repetitions (i.e., 33% intensity of effort) with 45 s of inter-set recovery, 6 sets of 6 repetitions (50% intensity of effort) with 72 s, and 4 sets of 9 repetitions (75% intensity of effort) with 120 s; with 4 min rest between exercises. Before and immediately after each session, arterial stiffness, hemodynamic variables, cardiac autonomic modulation, baroreflex sensitivity, sympathetic vasomotor tone, and resting oxygen uptake will be evaluated. Furthermore, perceived effort, mechanical performance (e.g., power, velocity), heart rate, and lactatemia will be collected throughout sessions. The impact of set configuration on these variables will be analyzed, along with comparisons between normotensive and hypertensive women.

**Data Availability Statement:** The original contributions of the study will be publicly available. These data will be stored in a European data repository (i.e., Zenodo) upon project completion, (https://zenodo.org/communities/udc/?page=1&size=20&size=20).

**Funding:** This work is part of the project supported by the Ministerio de Ciencia e Innovación/Agencia Estatal de Investigación of the Government of Spain. Grant PID2021-124277OB-I00 funded by MCIN/AEI/ 10.13039/501100011033 and, by "ERDF A way of making Europe". JR-V and MR-A acknowledge the financial support received from the Spanish Ministry of Universities through the Grants for the Requalification of the Spanish University System under the Postdoctoral Margarita Salas Program (JR-V: RSU.UDC.MS10; and MR-A: RSUC.UDC.MS09), funded by the European Union – Next Generation. The funders had no role in study design, data collection and analysis, decision to publish, or preparation of the manuscript.

**Competing interests:** The authors declare that the research was conducted in the absence of any commercial or financial relationships that could be construed as a potential conflict of interest.

## Discussion

Cardiovascular responses to resistance exercise have been scarcely studied in females, particularly postmenopausal women. The results of this study will provide information about the modulating role of set structure on metabolic and cardiovascular responses of normotensive and hypertensive postmenopausal women to resistance training.

## Clinical trial registration

NCT05544357 on 7 December 2022.

## Introduction

Hormonal changes of menopause have a detrimental effect on the health status of women, being associated with, among others, a decrease in bone mineral density, an increase in abdominal adiposity, a decrease in insulin sensitivity or higher susceptibility to cardiovascular disease (CVD) [1, 2]. For example, the prevalence of hypertension increases by up to 19% in premenopausal women, 44% in perimenopausal women, and 75% in postmenopausal women aged between 65 and 74 years old [3].

A substantial body of scientific-based recommendations in cardiovascular health have focused on promoting aerobic exercise due to its association with reduced CVD morbidity and all-cause mortality risk [4, 5]. However, emerging evidence highlights the benefits of strength training in reducing the risk of disease and mortality [6, 7]. In this regard, recent meta-analyses have shown the effectiveness of different types of resistance training to reduce resting diastolic and systolic blood pressure in both the general population [8] and specifically also in postmenopausal women with hypertension [9].

On the other hand, the estrogen deficiency has been associated with the accelerated loss in muscle mass and force observed in postmenopausal women in comparison with age-matched men, and with an impairment in the adaptation and recovery capacity of muscle, being these effects specific and independent of age [2, 10, 11]. In light of all these findings, resistance training emerges as a useful intervention to counteract the deleterious effects of hormonal changes in postmenopausal women [1, 2, 10].

Despite the acute and chronic benefits derived from muscle strength development programs, caution is warranted when implementing them in certain populations due to the pronounced hemodynamic response during resistance exercise with heavy loads [12]. This cardiovascular response includes a continuous increase in both systolic blood pressure (SBP) and diastolic blood pressure (DBP), heart rate, and the double product (i.e., Heat rate x SBP). Nevertheless, the duration of the exercise set is the primary factor influencing this elevation, with low-intensity sets with a high number of repetitions and close to muscular failure producing greater cardiovascular response [13, 14]. Nevertheless, in the prescription of exercise for individuals with cardiovascular diseases, the recommendation of using light weights and high repetitions still predominates [15]. This approach may limit strength development and increases cardiovascular stress, posing risks during strength training in a population with CVD risk. Hence, there is a need to design intervention approaches that combine improved mechanical stimulus quality (i.e., moderate to heavy loads) with reduced cardiovascular responses during exercise. An alternative set configuration design, different from the traditional model involving continuous repetitions until near muscular failure, involves modifying

work and rest intervals by breaking the series into small clusters of repetitions. This type of training has been termed cluster training, inter-repetition rest or intra-set rest training [16]. Previous studies have shown that shorter cluster-type series configurations reduce perceived effort [17, 18] and enhance mechanical performance during sessions in terms of speed and power with a lower glycolytic demand [19–21]. Additionally, recent studies have indicated that, compared to traditional configurations, relatively short set configurations (around 30–40% of maximum number of repetitions) attenuate hemodynamic responses and acute decrease of cardiac parasympathetic modulation [22–25]. In this regard, cluster-type set configurations have been recently recommended for prescribing safe resistance training in cardiac patients [26]. However, most of the studies about cardiovascular responses to different set configurations have been conducted on samples composed of young, healthy, physically active, and predominantly male individuals [22, 23, 25, 26]. To the best of our knowledge, the acute cardiovascular, autonomic, and metabolic responses to resistance training sessions differing in set configuration have not been explored yet in postmenopausal women. Given the deleterious effect of menopause on cardiovascular function [27–30] and muscle adaptation and recovery capacity [2, 11], we hypothesize that short-set structures could be particularly beneficial for this population as they may offer the benefits of strength training with lower cardiovascular risk and with lower perceived effort during exercise which might promote a higher adherence to training programs.

Taking into account all the previous considerations, this manuscript describes the protocol designed to study cardiovascular, neuromuscular and metabolic acute responses of physically active postmenopausal women to resistance training sessions differing in set configuration. Furthermore, this study aims to compare these responses between normotensive and hypertensive postmenopausal women. To mitigate the potential bias of elevated blood pressure solely due to sedentary lifestyles and considering that exercise training strategies tend to be more effective in individuals with higher baseline blood pressure [8, 31] the sample for this study will include normotensive and hypertensive samples for this study will be composed of physically active women.

## Materials and methods

### Experimental approach

The present study protocol was carried out in accordance with the SPIRIT (Standard Protocol Items: Recommendations for Intervention Data) guidelines [32] (Fig 1). The data collection and protocol intervention will be conducted in the facilities of the Faculty of Sports Sciences and Physical Education of the University of A Coruña (Spain). This study involves a randomized crossover design to examine the acute responses to resistance training sessions of different set configurations (S1 Fig). Each participant will complete one session oriented to medical evaluation, one familiarization session, two testing sessions to determine the 12 Repetitions Maximum load (12RM) of each exercise and their reliability, and four experimental sessions. Specifically, the experimental sessions will consist of three resistance training sessions with a different set configuration and one control session. The order of experimental sessions will be randomized. Briefly, the twenty-four possible sequences corresponding to four conditions will be numbered, and subsequently assigned to each participant through the generation of random numbers produced with a spreadsheet. The randomization will be performed without any influence from the principal investigators. During the experimental sessions, participants will perform the same volume (i.e., reps x kg) of the following resistance exercises: leg press, bench press, prone leg curl, and lat pull-down. Each session will comprise a different set configuration structure to maintain the same volume: 9 sets of 4 repetitions (4S), 6 sets of 6

| STUDY PERIOD | | | | | | | | |
| --- | --- | --- | --- | --- | --- | --- | --- | --- |
| | Enrollment | Allocation | Post-allocation | | | | | |
| TIMEPOINT | $-t_1$ | 0 | 12RM (1) | 12RM (2) | $S_1$ (+ 48-72h) | $S_2$ (+ 48-72h) | $S_3$ (+ 48-72h) | $S_3$ (+ 48-72h) |
| **ENROLLMENT:** | | | | | | | | |
| *Eligibility screen* | X | | | | | | | |
| *Informed consent* | X | | | | | | | |
| *Preliminary medical exam* | X | | | | | | | |
| *Allocation* | | X | | | | | | |
| *Pre-evaluation assessments* | X | | X | X | | | | |
| **INTERVENTIONS:** | | | | | *Randomized Order* | | | |
| *4S Set Configuration* | | | | | ◆━━━━━━━━━━━━━━━━━◆ | | | |
| *6S Set Configuration* | | | | | ◆━━━━━━━━━━━━━━━━━◆ | | | |
| *9S Set Configuration* | | | | | ◆━━━━━━━━━━━━━━━━━◆ | | | |
| *Control Session (CS)* | | | | | ◆━━━━━━━━━━━━━━━━━◆ | | | |
| **ASSESSMENTS:** | | | | | | | | |
| *Cardiopulmonary test* | X | | | | | | | |
| *Anthropometric & Body Composition* | X | | | | | | | |
| *Strength Performance* | | | X | X | | | | |
| *Arterial Stiffness* | | | | | X | X | X | X |
| *Hemodynamic* | | | | | X | X | X | X |
| *Cardiac Autonomic Control* | | | | | X | X | X | X |
| *Cardiac Baroreflex* | | | | | X | X | X | X |
| *Sympathetic Vasomotor Tone* | | | | | X | X | X | X |
| *Metabolic Responses* | | | | | X | X | X | X |
| *Mechanical Performance* | | | | | X | X | X | X |

**Fig 1. SPIRIT schedule of enrollment, interventions, and assessments.** 4S: 9 sets of 4 repetitions and 45s rest; 6S: 6 sets of 6 repetitions and 72s rest; 9S: 4 sets of 9 repetitions and 120s rest; CS: control session; 12RM: 12 Repetitions Maximum load test; $S_\#$: number of sessions.

repetitions (6S), and 4 series of 9 repetitions (9S). To ensure an equivalent work-to-rest ratio, the inter-set rest periods will be 45, 72, and 120 s for 4S, 6S, and 9S, respectively. In addition, the recovery time between exercises will be 4 min. Additionally, in order to obtain reference variables without the influence of resistance training session, a control session will be conducted, in which participants will be instructed to remain seated for 45 min. In addition, all experimental sessions will be separated by at least 48–72 h of recovery. Finally, in order to minimize the effects of circadian rhythm on collected variables, all sessions will be performed at the same time of the day (± 1 h).

## Ethical approval and registration

The design of this study was approved by the Galician Regional Government Ethical Committee (code: 2022/313) and is in accordance with principles outlined in the Declaration of Helsinki. Before signing an informed consent form, all participants will receive oral and written information about the study's aim, testing procedures, experimental sessions, potential benefits, and risks of their participation from the two project supervisors (EI-S and JF). Specifically, the research protocol was registered in the U.S. National Library of Medicine (ClinicalTrials. gov) receiving the code identification of NCT05544357 on 7th December 2022.

## Participants

A priori power analysis was conducted using G $^*$ Power V3 (Universität Kiel, Düsseldorf, Germany). Considering the research design, the sample size calculation was carried out for a statistical power of 80% (1– β) and assumed Type Error I of 0.05 in order to detect a small effect size (f = 0.12) for the interaction between the within-subject factor (i.e., set configuration 1–3 and control session) and between-subjects factor (i.e., normotensive and hypertensive groups) in an Analysis of Variance test (ANOVA), and assuming a correlation of 0.75 between repeated measures. As a result of this analysis, a total sample size of 25 women per group (i.e., a total sample of 50 participants) would be sufficient to detect small set configuration × group interaction effects. To avoid sample size problems related to potential dropouts (15% dropout rate), we will enroll a total of 60 women: 30 normotensive and 30 hypertensives. Regarding the recruitment procedure, the postmenopausal women will be recruited in sports facilities located in the city of A Coruña area. Furthermore, press releases and a profile on the main social media platforms (i.e., X and Instagram) will be created with specific details of the research design. Recruitment began on February 3, 2023. We will select physically active women to prevent the potential overlap of sedentary effects with the specific features of responses associated with menopause. The participants will have to fulfill the following inclusion criteria: (a) at least one year since the last menstrual period, (b) physically active (150–300 min weekly moderate physical activity or at least 75 min weekly vigorous activity), (c) $\leq$ 3 traditional cardiovascular risk factors, (d) asymptomatic and without cardiovascular (except hypertension), metabolic or renal diseases, and (e) in the case of hypertensive sample it must be composed of women diagnosed with well-controlled grade 1 hypertension, using a single drug. In addition, the exclusion criteria will be: (a) a diagnosis of grade 2 or 3 of hypertension, (b) hypertension controlled by more than one drug or by a medication that could potentially interfere with the cardiovascular responses to exercise (i.e., beta-blockers), (c) currently be receiving or have previously received hormonal replacement therapy, and (d) to show hypertensive response to exercise. Furthermore, to participate in the experimental research design, all participants will undergo a preliminary medical consultation to evaluate the absolute and relative contraindications for exercise and the inclusion and exclusion criteria. Additionally, in this medical exam, an

ergometry will be carried out following the Bruce protocol. The results of this test will be used to characterize the participants.

## Intervention

**Familiarization session.**   This session will be used to acquaint the participants with the proper execution of selected resistance exercises: horizontal leg press, bench press, prone leg curl and lat pull-down. Additionally, anthropometric profile and bone mineral density data recorded by an ultrasound bone mineral densitometer (Sonost 3000, Osteosys Corp., Korea) will be obtained. In order to standardize and ensure an optimal range of motion for each exercise, the individual marks will be registered to adjust the exercise machines in the subsequent sessions (i.e., pretesting and experimental sessions).

**Preliminary sessions: 12 RM test.**   Two pretesting sessions will be conducted to perform the 12RM test. In this respect, the 12RM test aims to determine the maximum load that each participant can lift 12 but not 13 times. As a specific warm-up, participants will be instructed to perform two sets of 12 repetitions with 2 min of rest between, corresponding to a perceived exertion (OMNI-RES scale) of 6, and 8 on a scale between 0 and 10 and the load will be increased accordingly. Then, participants will attempt the first evaluation of 12RM. If they perform 13 repetitions in the first set, load will be increased accordingly, and after 6 min of rest a new attempt will be carried out. Muscle failure will be identified when the participant is unable to overcome the load or when the full range of movement of the exercise is not completed.

**Experimental sessions.**   Before each experimental and the control session, all participants will be required to accomplish the following guidelines: (a) to maintain a minimum fasting period of 2 hours, (b) to avoid consuming caffeinated beverages, alcohol, or drugs 48 hours prior to sessions, and (c) not to perform strenuous exercise within 24 hours before each session.

Each session will start with 15 min of metabolic and cardiovascular evaluation at rest. For this assessment, participants will be placed supine on a stretcher with a 45º trunk inclination and will remain silent and avoid any movement for a 15 min period. Immediately after, an evaluation of arterial stiffness will be recorded in a horizontal lying position. Then participants will conduct a standardized warm-up involving: 5 min of cycling at a cadence of 60–80 revolutions per min, 2 min of joint mobility exercises, and a specific warm-up for 4 resistance exercises consisting of 10 repetitions with 70% of 12RM load. Afterward, participants will carry out the experimental sessions, in which exercises will be performed in the following order: (a) horizontal leg press (Sportsart N956, Switzerland), (b) bench press (Multipower Shock [Model SH004/0], Telju Fitness, Toledo, Spain), (c) prone leg curl (Biotech Fitness Solutions, Brazil) y (d) lat pull-down (Biotech Fitness Solutions, Brazil). The experimental sessions will be performed with the 12RM load previously determined. All resistance training sessions (i.e., 4S, 6S, and 9S) will comprise a total of 36 repetitions and 360 s of rest for each exercise. Each session will consist of a different set configuration structure ensuring that load, volume, rest interval, and session total duration are equated between conditions. In this respect, the 4S will include 9 sets of 4 repetitions (i.e., 33% of the intensity of effort) with 45 s of inter-set recovery for each exercise. In 6S, 6 sets of 6 repetitions (i.e., 50% of the intensity of effort) with 72 s of inter-set recovery will be performed for each exercise. Regarding 9S, 4 sets of 9 repetitions (i.e., 75% of the intensity of effort) of each exercise will be performed with an inter-set rest of 120 s. Previous studies have shown that these set structures cause different levels of cardiovascular, neuromuscular, and metabolic acute responses [22–24]. After each set, the perceived exertion will be recorded by supervisors using the OMNI-RES scale. For all resistance training sessions, a 4-min recovery will be established between exercises. Across all experimental sessions, heart

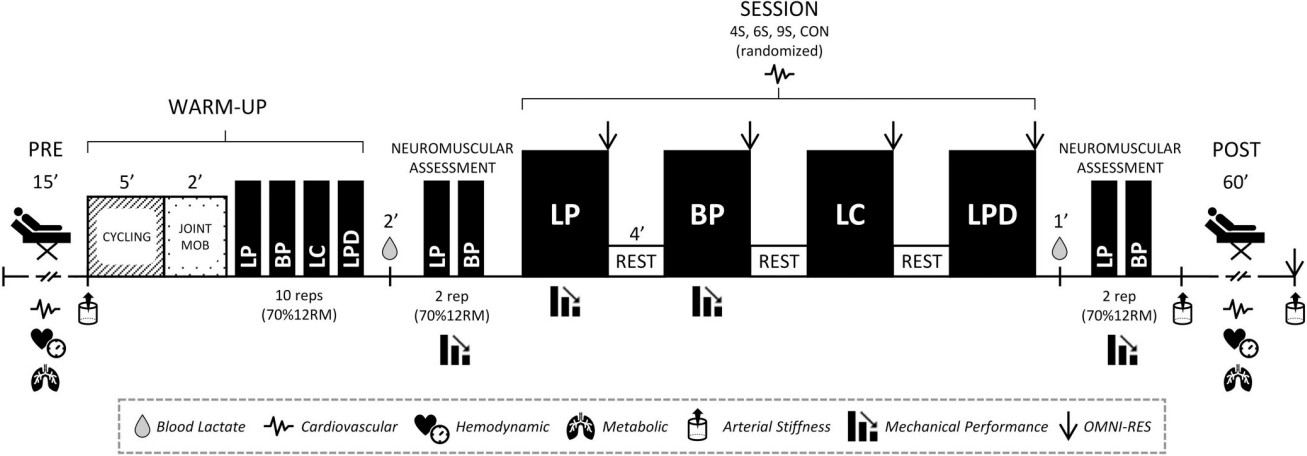

**Fig 2. Schematic representation of the experimental sessions.** 4S: 9 sets of 4 repetitions and 45s rest; 6S: 6 sets of 6 repetitions and 72s rest; 9S: 4 sets of 9 repetitions and 120s rest; CS: control session; 12RM: 12 Repetitions maximum load.

rate will also be monitored. Additionally, two min after the warm-up and one min after the last repetition of each session, capillary blood lactate recordings will be obtained in a seated position in order to estimate the glycolytic demand. Moreover, to evaluate neuromuscular fatigue, mean propulsive velocity against 70% of 12RM load for leg press and bench press will be measured after the warm-up and at the end of the session. Furthermore, mechanical performance in leg press and bench press exercises will be recorded throughout the experimental sessions.

Once finished exercise or control sessions, a post-test metabolic and cardiovascular evaluation at rest will be recorded for one hour following the procedures previously explained, except for arterial stiffness. Regarding this variable, a recording will be performed immediately after the protocol session and at the end of the one-hour period, followed by a request for the global perceived exertion.

All the intervention sessions will be supervised by the project leader (E.I.S.) and sports science researchers with experience in resistance training (J.R.V., J.F., and I.N.S.). Regarding resistance exercise execution, the supervisors will provide instructions to the participants to perform an eccentric controlled phase, while the concentric phase will be executed with the maximal intended velocity for each repetition. Fig 2 shows a schematic representation of experimental.

## Outcomes

Throughout all the experimental sessions, the outcomes measurement will present the same structure. In this regard, Table 1 summarizes primary and secondary outcomes that will be recorded in this study. Prior to each experimental session (15 min) and after sessions (60 min) the following outcomes will be collected: (a) arterial stiffness, (b) hemodynamic variables, (c) cardiac autonomic modulation (d) cardiac baroreflex, (e) sympathetic vasomotor tone, and (f) resting oxygen uptake.

During exercise, several fatigue indicators will be collected in order to characterize the sessions: (a) mechanical performance, and (b) cardiometabolic responses.

**Primary outcomes.** The primary outcomes include the cardiovascular and metabolic responses (Table 1). For cardiovascular outcomes, arterial stiffness will be measured by

**Table 1. Summary of outcomes and time points of measurements.**

| | PRIMARY OUTCOMES | | | |
|---|---|---|---|---|
| **Outcome** | | **Parameter** | **Description** | **Moment of measurement** |
| | | | **CARDIOVASCULAR OUTCOMES** | |
| **Arterial stiffness** | | **f-t PWV** | Finger-toe pulse wave velocity (m·s$^{-1}$) | Pre-Post session |
| **Hemodynamic** | | **SBP** | Systolic blood pressure (mmHg) | Pre-Post session (Epochs of 10 min) |
| | | **DBP** | Diastolic blood pressure (mmHg) | |
| | | **MAP** | Mean arterial pressure (mmHg) | |
| | | **HR** | Heart rate (bpm) | |
| | | **RPP** | Rate pressure product (bpm·mmHg) | |
| **Cardiac autonomic control** | **Time domain** | **SDNN** | Standard deviation of all NN intervals (ms) | Pre-Post session (Epochs of 5 min) |
| | | **RMSSD** | Square root of the mean of the sum of the squares of differences between adjacent NN intervals (ms) | |
| | **Frequency domain** | **LF** | Power of the low frequency range in absolute values (ms$^2$) | |
| | | **HF** | Power of the high frequency range in absolute values (ms$^2$) | |
| | | **RATIO LF_HF** | Ratio | |
| | **Entropy** | **SampEn** | Sample entropy | |
| **Cardiac baroreflex** | | **BRS$_{slope}$** | Magnitude of the baroreflex sensitivity (ms·mmHg$^{-1}$) | Pre-Post session (Epochs of 10 min) |
| **Sympathetic vasomotor tone** | | **LFsBP** | Low frequency of systolic blood pressure (mmHg$^2$) | |
| | | | **RESTING METABOLIC OUTCOMES** | |
| **Metabolic Responses** | | **VO$_2$** | Oxygen uptake (ml·kg$^{-1}$·min$^{-1}$) | Pre-Post session (Epochs of 10 min) |
| | | **VCO$_2$** | Carbon dioxide production (ml·kg$^{-1}$·min$^{-1}$) | |
| | | **RER** | Respiratory exchange ratio | |
| | SECONDARY OUTCOMES | | | |
| | | | **CARDIOVASCULAR OUTCOMES** | |
| **Arterial Stiffness** | | **f-t TT** | Finger-toe transit time (ms) | Pre-Post session |
| **Cardiac autonomic control** | **Frequency domain** | **LF n.u.** | Power of the low frequency range in normalized units | Pre-Post session (Epochs of 5 min) |
| | | **HF n.u.** | Power of the high frequency range in normalized units | |
| | **Entropy** | **AmpEn** | Approximate entropy | |
| **Cardiac baroreflex** | | **BRS$_{count}$** | Number of baroreflex sequences detected (n) | Pre-Post session (Epochs of 10 min) |
| | | **BEI** | Baroreflex effectiveness index (%) | |
| | | | **NEUROMUSCULAR FATIGUE OUTCOMES** | |
| **Mechanical performance** | | **VMP-70%** | Mean propulsive velocity against 70% of 12RM load (m·s$^{-1}$) | During sessions |
| | | **VM** | Mean velocity (m·s$^{-1}$) | |
| | | **PM** | Mean power (m·s$^{-1}$) | |
| | | **FM** | Mean force (N) | |
| | | | **CARDIOMETABOLIC OUTCOMES** | |
| **Cardiometabolic responses** | | **La** | Capillary blood lactate (mmol·L$^{-1}$) | Pre-Post session |
| | | **HR** | Heart rate (bpm) | During sessions |
| | | **RPE** | Rate of perceived exertion | |
| | | | **CARDIOPULMONARY TEST** | |
| **Cardiovascular responses to effort test** | | **VO$_{2max}$** | Maximum oxygen uptake (ml·kg$^{-1}$·min$^{-1}$) | During effort test |
| | | **HR$_{max}$** | Maximum heart rate during effort test (bpm) | |
| | | **SBP$_{max}$** | Maximum systolic blood pressure during effort test (mmHg) | |
| | | **DBP$_{max}$** | Diastolic blood pressure during effort test (mmHg) | |

(*Continued*)

**Table 1.** (Continued)

| ANTROPOMETRIC AND BODY COMPOSITION | | | |
|---|---|---|---|
| Anthropometric description | Weight | (kg) | Recorded in pre-evaluation assessments |
| | Height | (cm) | |
| | BMI | Body mass index [weight(kg); (height(m)$^2$)] kg·m$^{-2}$ | |
| | % fat mass | (%) | |
| | % fat free mass | (%) | |
| | SOS | Speed of sound | |
| | BUA | Broadband ultrasound attenuation | |
| | BQI | Bone quality index | |
| PHYSICAL FITNESS- STRENGTH COMPONENT | | | |
| Strength performance | 12RM | 12 repetitions maximum load (kg) for each exercise (i.e., horizontal leg press, bench press, leg curl, lat pull-down) | Recorded in pre-evaluation assessments |

pOpmètre (Axelife sas, France) with pOplog 3.1.67 software [33]. This evaluation requires prior blood pressure measurement, obtained by an oscillometric device (Omron MIT Elite Plus, Kyoto, Japan) with the cuff located in the left arm whereas cardiac autonomic control, cardiac baroreflex, sympathetic vasomotor tone, and hemodynamics will be assessed with the Task Force® Monitor (CNSystems, Graz, Austria) and TFM software v2.3 (CNSystems, Graz, Austria) [34]. For resting metabolic outcomes, a portable gas analyzer (MetaMax 3b; Cortex Biophysik, Leipzig, Germany) and the specific capture software (MetaSoft® Studio v5.13.0) will be used [35].

**Secondary outcomes.** The secondary outcomes will be the neuromuscular fatigue assessed by mechanical performance using a linear velocity transducer (T-Force System; Ergotech, Murcia, Spain) with a specific software (T-Force Dynamic Measurement System v3.70), glycolytic demand of the sessions estimated by capillary blood lactate measurements (Lactate Pro 2; Arkray, Kioto, Japan), perceived exertion after during session and at the end of the session with the OMNI-RES scale, and heart rate. The latter will be registered using a beat-to-beat heart rate sensor (Polar H10 model; Polar Electro Oy, Kempele, Finland) recorded using a mobile application (Elite HRV Inc., Asheville, NC, USA) and processed using Kubios HRV v.3.5.0 software (Kubios, Ltd., Kuopio, Finland) [36].

Additionally, anthropometric characteristics, body composition and physical fitness of the sample will be recorded. For anthropometric and body composition parameters, a stadiometer (Seca 202; Seca Ltd., Hamburg, Germany), a quadruple multifrequency bioelectrical impedance device (Omron BF-508; Omron Healthcare Co., Kyoto, Japan), and an ultrasound bone mineral densitometer Sonost 3000 (Osteosys Corp., Korea) will be used. For physical fitness parameters, the cardiorespiratory component will be evaluated by the Bruce protocol, in which a portable gas analyzer (MetaMax 3b; Cortex Biophysik, Leipzig, Germany), a heart rate monitor (Polar H10 model; Polar Electro Oy, Kempele, Finland), sphygmomanometer (Riester, Jungingen, Germany), and a phonendoscope (3M Health Care, ST. Paul, MN, USA) will be used. For the muscular component, strength of participants will be characterized by the 12 RM load of each exercise.

## Control and management of adverse effects

In this research design, the measurements and procedures will present a low risk of adverse effects due to being minimally invasive. In this sense, the preliminary and experimental

sessions could lead to acute increases in blood pressure, muscle pain, and delayed onset muscle soreness [37, 38]. In addition, all the participants could be at minimal risk of musculoskeletal injuries during resistance training sessions. In order to control and minimize these possible adverse effects, a physician and a group of experienced researchers in resistance training will supervise all procedures and training sessions. If any of the potential or chronic exercise-related adverse effects previously mentioned were identified, the physician of the study will perform a participant's health examination. Finally, if the study physician considered that the participant's health conditions may represent a potential risk, decisions regarding future participation in the current protocol will be made in consensus with the participant.

## Statistical analyses

Descriptive statistics will be reported as means and standard deviation (mean ± SD). All statistical analyses will be conducted using the statistical package of SPSS version 25.0 (SPSS, IBM, Armonk, NY, USA) and R version 4.2.1 (R Foundation, Vienna, Austria). For each variable, the Shapiro-Wilk test will be used in order to check the normality of the data distribution, and the homogeneity will be examined using Levene's Test. If data present a normal distribution, a two-way repeated-measures ANOVA will be conducted. Specifically, a 4 x 2 repeated-measures ANOVA will be performed to analyze the effects of set configuration (i.e., CON, 4S, 6S, and 9S) across the time (i.e., pre-test vs. post-test). If a significant interaction is detected, pairwise comparisons using the Bonferroni post-hoc test will be conducted to examine the statistical differences between set configurations and times. Additionally, to address missing or incomplete values in any participant and to facilitate the inclusion of fixed and random effects factors as covariates or model factors, we will use linear mixed models. Conversely, in cases where the assumption of normality is violated, nonparametric ANOVA-type statistics will be performed [39]. This nonparametric analysis provides a similar analysis compared to parametric ANOVA but uses rank-based calculations to determine the relative marginal effects [39]. If a significant interaction is found, paired comparisons within-group will be conducted using the Wilcoxon signed-rank with Bonferroni's adjustment. For all statistical analyses, the significance will be established at the $P \leq 0.05$ levels.

To assess the magnitude of differences for parametric variables, two effect size measures will be employed: partial eta squared ($\eta p2$) and Hedge's G with 95% of confidence intervals. Additionally, the matched-pair rank-biserial correlation will be implemented to calculate the effect size in nonparametric contrasts.

## Discussion

Changes associated with menopause have a deleterious effect on the metabolic and cardiovascular health of women [1, 2, 10, 11]. Furthermore, hormonal changes in postmenopausal women negatively affect muscle composition, quality, and capacity to recover and adapt from exercise [11]. Thus, resistance training is recommended to counteract detrimental health changes during menopause [1, 2, 10].

In this regard, the positive effect of resistance training on health has been extensively shown, including structural improvements (preventing sarcopenia and osteoporosis), functional enhancements (mobility, activities of daily living, fall prevention), cognitive and mental health, metabolic (glycated hemoglobin, metabolism, lipid profiles) and cardiovascular health benefits (blood pressure reduction, improved autonomic control) [40, 41]. Overall, muscle strength activities have been inversely associated with the risk of all-cause mortality and major non-communicable diseases [6].

However, there are still concerns regarding the utilization of resistance training in individuals with cardiovascular risk, primarily due to the abrupt hemodynamic acute responses associated with this form of exercise [12]. In this context, duration of exercise (i.e., time and number of repetitions) and proximity to muscular failure, known as intensity of effort or level of effort, have been recognized as key factors for regulating the cardiovascular response to resistance exercise [13, 14].

Manipulating set configurations is a means to modulate the level of effort and, therefore the cardiovascular responses. Previous studies have shown that performing a low number of repetitions per set (i.e., short set configurations) allows attenuating hemodynamic responses and acute changes in cardiac autonomic control in comparison with training routines of traditional set configurations (i.e., long set configurations performed close to muscular failure) even when volume and work-to-rest ratio were equated [22–25]. However, to the best of our knowledge, the modulator effect of set configuration on hemodynamic and cardiac autonomic control has not been previously and specifically explored in normotensive and hypertensive postmenopausal women.

The study aims to compare different set configurations of resistance training in terms of their impact on mechanical, metabolic, and cardiovascular acute responses in postmenopausal women. Furthermore, the study will investigate disparities in these acute adaptations between normotensive and hypertensive participants. Based on previous findings, we hypothesize: (i) a lower acute reduction of parasympathetic and baroreflex cardiac control after sessions with short set configurations (i.e., 4S and 6S) in comparison with those carried out with long set configurations (i.e., 9S); (ii) a more pronounced hypotensive effect of resistance training sessions after long set configurations and particularly in hypertensive participants; and (iii) a higher resting oxygen uptake increase after sessions with long set configurations.

The results of this study will provide relevant information to identify the optimal set structure to be used with postmenopausal women according to the intended objective of the session. Thus, the findings of this project will allow to select the suitable set configurations to limit or increase in post-menopausal women the acute mechanical, metabolic and cardiovascular responses to resistance training sessions. However, some limitations of the design must be considered. First, the implications of differences in acute responses on long-term effects are beyond the scope of this design, and further studies will be needed to address this matter. For example, one may inquire whether increased acute cardiovascular stress does or does not lead to greater adaptation in the medium term. Secondly, our study will be performed with physically active postmenopausal women to avoid the interaction between menopause and sedentarism. Translating the results to women with low levels of physical activity should be carried out with caution.

## Dissemination

This study aims to provide objective data on the effects of resistance training on postmenopausal women. The information obtained will enable safe and appropriate prescription of this type of exercise for this population. The results will be actively disseminated through peer-reviewed journals, conference presentations, social media, and community engagement activities, as well as to study participants.

## Supporting information

**S1 Fig. Flowchart of the study protocol implementation.** 4S: 9 sets of 4 repetitions and 45s rest; 6S: 6 sets of 6 repetitions and 72s rest; 9S: 4 sets of 9 repetitions and 120s rest; CS: control

session.
(TIF)

**S1 Checklist. SPIRIT checklist.**
(DOC)

**S1 Protocol. Trial Protocol study_Spanish version.**
(DOCX)

**S2 Protocol. Trial Protocol study_English version.**
(DOCX)

## Author Contributions

**Conceptualization:** Alexis Padrón-Cabo, María Rúa-Alonso, Manuel Giráldez-García, Eliseo Iglesias-Soler.

**Funding acquisition:** Eliseo Iglesias-Soler.

**Project administration:** Eliseo Iglesias-Soler.

**Supervision:** Iván Nine, Jessica Rial-Vázquez, Juan Fariñas.

**Writing – original draft:** Iván Nine, Alexis Padrón-Cabo, Eduardo Carballeira, Jessica Rial-Vázquez, María Rúa-Alonso, Manuel Giráldez-García, Eliseo Iglesias-Soler.

**Writing – review & editing:** Iván Nine, Alexis Padrón-Cabo, Jessica Rial-Vázquez, María Rúa-Alonso, Juan Fariñas, Eliseo Iglesias-Soler.

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
