## [Decision Letter · Decision Letter 0]

26 Aug 2024

PONE-D-24-28700Acute cardiovascular responses of postmenopausal women to resistance training sessions differing in set configuration: a study protocol for a crossover trialPLOS ONE

Dear Dr. Iglesias-Soler,

Thank you for submitting your manuscript to PLOS ONE. After careful consideration, we feel that it has merit but does not fully meet PLOS ONE’s publication criteria as it currently stands. Therefore, we invite you to submit a revised version of the manuscript that addresses the points raised during the review process.

We look forward to receiving your revised manuscript.

Kind regards,

Daniel Boullosa

Academic Editor

PLOS ONE

Journal Requirements:

When submitting your revision, we need you to address these additional requirements. 1. Please ensure that your manuscript meets PLOS ONE's style requirements, including those for file naming. The PLOS ONE style templates can be found at https://journals.plos.org/plosone/s/file?id=wjVg/PLOSOne_formatting_sample_main_body.pdf and https://journals.plos.org/plosone/s/file?id=ba62/PLOSOne_formatting_sample_title_authors_affiliations.pdf 2. Thank you for stating the following financial disclosure: "This work is part of the project PID2021-124277OB-I00 funded by MCIN/AEI/10.13039/501100011033/FEDER, UE. JR-V and MR-A acknowledge the financial support received from the Spanish Ministry of Universities through the Grants for the Requalification of the Spanish University System under the Postdoctoral Margarita Salas Program (JR-V: RSU.UDC.MS10; and MR-A: RSUC.UDC.MS09), funded by the European Union – Next Generation." Please state what role the funders took in the study.  If the funders had no role, please state: ""The funders had no role in study design, data collection and analysis, decision to publish, or preparation of the manuscript."" If this statement is not correct you must amend it as needed. Please include this amended Role of Funder statement in your cover letter; we will change the online submission form on your behalf. 3. Your ethics statement should only appear in the Methods section of your manuscript. If your ethics statement is written in any section besides the Methods, please delete it from any other section.  4. We note that Figure 2 in your submission contain copyrighted images. All PLOS content is published under the Creative Commons Attribution License (CC BY 4.0), which means that the manuscript, images, and Supporting Information files will be freely available online, and any third party is permitted to access, download, copy, distribute, and use these materials in any way, even commercially, with proper attribution. For more information, see our copyright guidelines: http://journals.plos.org/plosone/s/licenses-and-copyright. We require you to either (1) present written permission from the copyright holder to publish these figures specifically under the CC BY 4.0 license, or (2) remove the figures from your submission: a. You may seek permission from the original copyright holder of Figure 2 to publish the content specifically under the CC BY 4.0 license.  We recommend that you contact the original copyright holder with the Content Permission Form (http://journals.plos.org/plosone/s/file?id=7c09/content-permission-form.pdf) and the following text:“I request permission for the open-access journal PLOS ONE to publish XXX under the Creative Commons Attribution License (CCAL) CC BY 4.0 (http://creativecommons.org/licenses/by/4.0/). Please be aware that this license allows unrestricted use and distribution, even commercially, by third parties. Please reply and provide explicit written permission to publish XXX under a CC BY license and complete the attached form.” Please upload the completed Content Permission Form or other proof of granted permissions as an ""Other"" file with your submission.  In the figure caption of the copyrighted figure, please include the following text: “Reprinted from [ref] under a CC BY license, with permission from [name of publisher], original copyright [original copyright year].” b. If you are unable to obtain permission from the original copyright holder to publish these figures under the CC BY 4.0 license or if the copyright holder’s requirements are incompatible with the CC BY 4.0 license, please either i) remove the figure or ii) supply a replacement figure that complies with the CC BY 4.0 license. Please check copyright information on all replacement figures and update the figure caption with source information. If applicable, please specify in the figure caption text when a figure is similar but not identical to the original image and is therefore for illustrative purposes only. 5. We are unable to open your Supporting Information file 
SUPPORTING INFORMATION.rar. Please kindly revise as necessary and re-upload.

Reviewers' comments:

Reviewer's Responses to Questions

**Comments to the Author**

1. Does the manuscript provide a valid rationale for the proposed study, with clearly identified and justified research questions?

Reviewer #1: Yes

Reviewer #2: Yes

2. Is the protocol technically sound and planned in a manner that will lead to a meaningful outcome and allow testing the stated hypotheses?

Reviewer #1: Yes

Reviewer #2: Yes

3. Is the methodology feasible and described in sufficient detail to allow the work to be replicable?

Reviewer #1: Yes

Reviewer #2: Yes

4. Have the authors described where all data underlying the findings will be made available when the study is complete?

Reviewer #1: Yes

Reviewer #2: No

5. Is the manuscript presented in an intelligible fashion and written in standard English?

Reviewer #1: Yes

Reviewer #2: Yes

6. Review Comments to the Author

You may also provide optional suggestions and comments to authors that they might find helpful in planning their study.

Reviewer #1: Very nice. Considering this is a protocol paper, I would say everything is well-described and the rationale is there. To me, those are the most important aspects for such a paper. Clearly, the "discussion" pointes are theoretical, but should be well-defined and connected to the current literature when the scientific publications come out. I look forward to the paper(s) in the future. Keep up the good work.

Reviewer #2: In this study protocol, a 4 segment randomized crossover design is being proposed to evaluate the impact on resistance training set configuration on arterial stiffness, hemodynamic variables, cardiac autonomic, modulation, baroreflex sensitivity, sympathetic vasomotor tone, resting oxygen uptake, perceived effort, mechanical performance, heart rate, and lactatemia, along with comparisons between the normotensive and hypertensive groups.

Minor revisions:

1- Line 180: State the statistical testing method which achieves 80% power.

2- Identify the software that will be used to capture the data.

7. PLOS authors have the option to publish the peer review history of their article (what does this mean?). If published, this will include your full peer review and any attached files.

Reviewer #1: No

Reviewer #2: No

---

## [Author Response · Author response to Decision Letter 0]

6 Sep 2024

RESPONSE: We have meticulously adhered to the template and ensured that all files are named in accordance with the correct format as outlined in the style guidelines.

"This work is part of the project PID2021-124277OB-I00 funded by MCIN/AEI/10.13039/501100011033/FEDER, UE. JR-V and MR-A acknowledge the financial support received from the Spanish Ministry of Universities through the Grants for the Requalification of the Spanish University System under the Postdoctoral Margarita Salas Program (JR-V: RSU.UDC.MS10; and MR-A: RSUC.UDC.MS09), funded by the European Union – Next Generation."

RESPONSE: The statement you suggested has been incorporated into the revised version of the cover letter.

RESPONSE: The Ethical Statement has been removed from other sections and now appears exclusively in the Methods section.

4. We note that Figure 2 in your submission contain copyrighted images. All PLOS content is published under the Creative Commons Attribution License (CC BY 4.0), which means that the manuscript, images, and Supporting Information files will be freely available online, and any third party is permitted to access, download, copy, distribute, and use these materials in any way, even commercially, with proper attribution. 

RESPONSE: We have removed the potentially copyrighted images and provided a new figure with images that we created ourselves and are free of copyright.

5. We are unable to open your Supporting Information file SUPPORTING INFORMATION.rar. Please kindly revise as necessary and re-upload.

RESPONSE: We regret this issue. We have now uploaded the Supporting Information as individual files to avoid any problems with the compressed folder.

RESPONSE: We have reviewed the reference list and confirmed that all cited papers are accurate, and none have been retracted.

Reviewers' comments:

Reviewer #1: Very nice. Considering this is a protocol paper, I would say everything is well-described and the rationale is there. To me, those are the most important aspects for such a paper. Clearly, the "discussion" pointes are theoretical, but should be well-defined and connected to the current literature when the scientific publications come out. I look forward to the paper(s) in the future. Keep up the good work.

RESPONSE: We sincerely appreciate the reviewer’s comments. Thank you very much for your valuable feedback

Reviewer #2: In this study protocol, a 4 segment randomized crossover design is being proposed to evaluate the impact on resistance training set configuration on arterial stiffness, hemodynamic variables, cardiac autonomic, modulation, baroreflex sensitivity, sympathetic vasomotor tone, resting oxygen uptake, perceived effort, mechanical performance, heart rate, and lactatemia, along with comparisons between the normotensive and hypertensive groups.

RESPONSE: We thank the reviewer for their comments and suggestions. Please find below our responses to your specific comments.

Minor revisions:

1- Line 180: State the statistical testing method which achieves 80% power.

RESPONSE: Thank you for your comment. We conducted two types of power calculations for ANOVA tests: one for the interaction in a 4 x 2 repeated-measures ANOVA (set configuration * time) and another for the interaction in a 4 x 2 ANOVA involving a within-subject factor (i.e., set configuration 1-3 and control session), and a between-subjects factor (i.e., hypertensive and normotensive). For the same conditions, the former analysis required 49 participants, while the latter needed a total sample of 50 (25 per group). Therefore, we decided to report the more demanding test. Additionally, as suggested by the reviewer, we have specified in the revised manuscript the test for which the sample size was calculated. “A priori power analysis was conducted using G * Power V3 (Universität Kiel, Düsseldorf, Germany). Considering the research design, the sample size calculation was carried out for a statistical power of 80% (1– β) and assumed Type Error I of 0.05 in order to detect a small effect size (f = 0.12) for the interaction between the within-subject factor (i.e., set configuration 1-3 and control session) and between-subjects factor (i.e., normotensive and hypertensive groups) in an Analysis of Variance test (ANOVA), and assuming a correlation of 0.75 between repeated measures”

2- Identify the software that will be used to capture the data.

RESPONSE: The software to be used for capturing each variable has been added as requested.

---

## [Decision Letter · Decision Letter 1]

20 Sep 2024

Acute cardiovascular responses of postmenopausal women to resistance training sessions differing in set configuration: a study protocol for a crossover trial

PONE-D-24-28700R1

Dear Dr. Iglesias-Soler,

We’re pleased to inform you that your manuscript has been judged scientifically suitable for publication and will be formally accepted for publication once it meets all outstanding technical requirements.

Kind regards,

Daniel Boullosa

Academic Editor

PLOS ONE

Additional Editor Comments (optional):

Reviewers' comments:

Reviewer's Responses to Questions

**Comments to the Author**

1. Does the manuscript provide a valid rationale for the proposed study, with clearly identified and justified research questions?

Reviewer #1: Yes

Reviewer #2: Yes

2. Is the protocol technically sound and planned in a manner that will lead to a meaningful outcome and allow testing the stated hypotheses?

Reviewer #1: Yes

Reviewer #2: Yes

3. Is the methodology feasible and described in sufficient detail to allow the work to be replicable?

Reviewer #1: Yes

Reviewer #2: Yes

4. Have the authors described where all data underlying the findings will be made available when the study is complete?

Reviewer #1: Yes

Reviewer #2: No

5. Is the manuscript presented in an intelligible fashion and written in standard English?

Reviewer #1: Yes

Reviewer #2: Yes

6. Review Comments to the Author

You may also provide optional suggestions and comments to authors that they might find helpful in planning their study.

Reviewer #1: None, accept as is…………………………………………………………………………………………………….. . Must have 100 characters. Sorry. …………….

Reviewer #2: All comments have been adequately addressed.

7. PLOS authors have the option to publish the peer review history of their article (what does this mean?). If published, this will include your full peer review and any attached files.

Reviewer #1: No

Reviewer #2: No

---

## [Editor Report · Acceptance letter]

4 Oct 2024

PONE-D-24-28700R1 

PLOS ONE

Dear Dr. Iglesias-Soler, 

I'm pleased to inform you that your manuscript has been deemed suitable for publication in PLOS ONE. Congratulations! Your manuscript is now being handed over to our production team.

Kind regards, 

on behalf of

Dr. Daniel Boullosa 

Academic Editor

PLOS ONE